# Coping with the Experience of Bad Sleep and Fatigue Associated with the Nursing Clinical Practicum

**DOI:** 10.3390/ijerph19127479

**Published:** 2022-06-18

**Authors:** Mei-Hsin Lai, Chyn-Yuan Tzeng, Yeu-Hui Chuang, Pi-Chen Chang, Min-Huey Chung

**Affiliations:** 1School of Nursing, College of Nursing, Taipei Medical University, 250 Wu-Hsing St., Taipei City 110, Taiwan; g8805001@sunrise.hk.edu.tw (M.-H.L.); yeuhui@tmu.edu.tw (Y.-H.C.); pichen@tmu.edu.tw (P.-C.C.); 2Department of Nursing, Hungkuang University, No. 1018, Sec. 6, Taiwan Boulevard, Shalu District, Taichung 433304, Taiwan; 3Taiwan Home Care & Service Association, Room A1415H, Medical Building, 250 Wu-Hsing St., Taipei City 110, Taiwan; tcy9907@gmail.com; 4Center for Nursing and Healthcare Research in Clinical Application, Wan Fang Hospital, Taipei Medical University, 111 Sec. 3. Xinglong Rd, Wenshan District, Taipei 116, Taiwan; 5Department of Nursing, Shuang Ho Hospital, Taipei Medical University, New Taipei City 23561, Taiwan

**Keywords:** grounded theory, coping, bad sleep, fatigue, practicum-related anxiety symptoms

## Abstract

Nursing students experience anxiety during clinical practicum, which may interfere with their learning of clinical practice. This study explored the practicum anxiety symptom experience of clinical nursing students. The methodology was mixed research design consist of questionnaire and a qualitative research design following a grounded theory approach. Research data were gathered via theoretical sampling from 37 female college nursing students practicing in a Central Taiwan hospital and analyzed using the content analysis method. The mean age of the subjects was 20.7 ± 1.35 years old. The practicum-related anxiety symptom experience was made up of core dimensions associated with the nursing clinical practicum, such as bad sleep and fatigue, and covered six themes. Students first encountered anxiety-inducing situations in the clinical setting, and then they began coping through self-adaptation and teachers’ help. In cases of coping failure, students began to have a bad sleep in the night and then felt tired and fatigued all day. These themes became a repeating cycle during the clinical practicum. This conceptual model shows that students experienced bad sleep and fatigue as a result of anxiety symptoms during the clinical practicum. Bad sleep and fatigue are critical anxiety symptoms for nursing students in clinical practice. Nursing teachers should pay attention to bad sleep and fatigue in nursing students and help students to improve their mental and physical health.

## 1. Introduction

Nursing is what kind of a profession? Warm? Cool? Gendered? Kind? The characteristics of a profession are usually dependent on the personality traits of its members [1,2]. Therefore, training a nursing student to be a professional nurse is an important issue. It means nursing students need special training or education to learn to be an autonomous nursing professional [3].

Anxiety is a condition that develops in nursing students undergoing training in clinical practice. This may be stimulated by clinical situations that lead to feelings of uncertainty. When practicing nursing care, anxiety-inducing situations might interrupt the student nurse’s inner learning motives, ability to assist, and level of hard work [4,5]. Many nursing studies mentioned their anxiety status, whether or not they were stimulated by the clinical practicum situation [6,7,8].

According to the Yerkes−Dodson Law, only when relaxing can people work efficiently and to a good standard, so students can only work well with adequate and suitable anxiety levels. Difficulty learning causes premature cognitive narrowing and this narrowing makes it difficult to resolve issues and learn well [9]. Self-compassion can indirectly influence anxiety through level of perceived stress [4]. The high levels of anxiety that nursing students experience may interfere with learning by impacting their thinking and performance ability during clinical practice [10,11]. Anxiety can also impact nursing students’ sleep. Dalcali et al. [12] mentioned that anxiety may interrupt nursing students’ sleep. Although the practicum is very important to nursing students, many of them still experience a state of anxiety upon participating in it regularly. Anxiety is an emerging study as the literature has reported the effects of anxiety status on student nurses, but there are few studies that explore the influence of consistent practicum anxiety on the system. 

Fatigue is a common complaint among medical students. Sometimes people express it through tiredness. Medical outcomes of fatigue include weakness (potentially associated with neuromuscular disorders) and sleepiness (for example, lethargy) [13]. The concept of fatigue influences three dimensions: physical, cognitive, and emotional state. Therefore, fatigued people would show symptoms such as a lack of energy, the need to take a rest, significantly decreased sensory focus, significantly decreased attention, and a decrease in motivation. Therefore, fatigue can affect the behavior of nurses and enhance risk in the workplace. Yiğitalp and Aydin [14] studied nursing students and mentioned that bad sleep and fatigue symptoms commonly occurred in nursing students. The severity of fatigue accompanied by low sleep quality increased and may affect the performance of students. Therefore, fatigue is a critical condition and an individual needs to have the ability to cope in a crisis so they are able to respond and cope appropriately to these critical situations [15].

Furthermore, regarding the issue of somatic symptoms during mental stress, a study carried out by Tian et al. [16] revealed that fatigue may occur due to stress and the fatigue may also further influence the stress. There was a high correlation and seemed to be a two-sided relationship between fatigue and perceived stress.

Mark et al.’s [17] study on the relationship between coping, anxiety, and fatigue, demonstrated that coping styles are related to stress and the coping styles are correlated with fatigue in medical students. Pourabdian et al.’s [15] study showed that fatigue could change one’s coping style from a problem-solving style to an emotional-oriented and avoidance style. Hence, subjects should use positive coping strategies to avoid fatigue. There were a range of strategies that enabled nursing students to positively cope with anxiety-inducing situations in clinical practice, such as debriefing clinical lecturers, seeking friend and family support, being adequately prepared, identifying and seeking advice from supportive nursing staff, and employing effective communication [18,19].

There have been many studies researching the areas of stress, fatigue, and coping. But results have shown a lack of a comprehensive understanding of the subjective experiences of anxiety in nursing students participating in clinical practicums among Taiwanese nursing students; in particular, there has been a lack of understanding of practicum-related anxiety symptoms. This requires further exploration as it is an important issue to nursing students and teachers. 

Given the lack of conclusive evidence in the available literature, we identified a need to understand practicum-related anxiety symptoms and their characteristics. 

The purpose of this study was to explore female student nurses’ experiences of practicum-related anxiety symptoms, aimed to establish a descriptive, conceptual model of the subjective experiences of practicum-related anxiety symptoms in Taiwanese female nursing students in the clinical practicum.

## 2. Materials and Methods

### 2.1. Study Design

This study used a mixed methods design to establish a comprehensive view of participants’ practicum-related anxiety symptoms and decrease the limitations of a single method design [20]. The design consisted of an anxiety inventory for quantitative analysis and an individual participant open questionnaire for qualitative analysis [21]. The qualitative research was designed according to Strauss and Corbin’s [22,23] grounded theory (GTM) approach, which provides a systematic guideline to gather varied forms of data, such as ethnographic field notes, written person accounts, and documents. The GTM provides authors with the ability to synthesize, analyze, and conceptualize qualitative data in the study to reach the goal of theory construction.

### 2.2. Participants

The participants of this study were comprised of 37 anxious female students from a total of 245 nursing students practicing in a hospital in Central Taiwan between January 2013 and December 2013. All the participants were female nursing students practicing in Kuang Tien General Hospital, one of the major practicum placements for nursing students in Central Taiwan. This is a regional teaching hospital in the central area of Taiwan, in the Shalu District of Taichung City in Taiwan, Republic of China (R. O. C.), involving a branch (Dajia Branch) in Dajia of Taichung City in Taiwan, Republic of China (R. O. C.). The distance from Shalu Kuang Tien General Hospital to the Dajia Branch is 30–40 min. In this study, we visited the nursing students practicing in these two hospitals.

### 2.3. Sample 

Theoretical sampling [24] was adopted in this study based on the following criteria: participants (1) had reached the end-week of the practicum period; (2) were female; (3) had no language disturbance that strained their understanding and writing abilities; (4) had a pre- or post-practicum BAI score ≥ 8 and post-practicum anxiety was higher than or equal to pre-practicum anxiety; and (5) were willing to share their clinical nursing practicum experiences and agreed to have the qualitative questionnaire analyzed.

### 2.4. Instrument

#### 2.4.1. Open-Ended Question Questionnaire

The authors developed an open-ended question questionnaire according to a grounded theory data collection method [23,25]. The open-ended questionnaire encouraged nursing students to write their own responses on their anxiety experience during clinical practice. 

The qualitative questionnaires were drafted according to the study purpose, allowing the answerers to write freely and to represent naturalistic [26] inquiry into their experience of anxiety to represent fact from the perspective of nursing students.

The qualitative post-practicum questionnaire used a research guide, which included two opening questions. The first question asked participants: “please describe your cur-rent anxiety events during nursing practicum”. Then, “please talk about your current stressful situations” was written, and a series of sub-questions followed: “please describe the anxiety events of difficulty in adapting to the medical team”, “please describe the anxiety events of difficulty in performing nursing processes”, and “please describe the anxiety events of difficulty in performing nursing skills.”

The second question was: “please describe your resources for current anxiety events during nursing practicum.” Then, “please talk about your supporters and what made working for current anxiety events in the practicum environments” was asked, and a series of sub-questions followed: “please describe the assistances from your school teacher in clinical practicum setting could be helpful to you (Please elaborate the reasons if any)?” and “please describe the assistances from your advisors in clinical practicum setting could be helpful to you (Please elaborate the reasons if any)”?

#### 2.4.2. Beck Anxiety Inventory

The quantitative questionnaire used at the pre- and post-practicum stage was the well-developed, translated-to-Chinese version, which possesses a high internal consistency (Cronbach’s α = 0.95, Guttman split−half coefficient = 0.91) and has been validated (two-factor structure such as Beck’s original construct and a total variance of 58.04% through a factor analysis) [27]. The Beck Anxiety Inventory (BAI) is a 21-item, multiple-choice and self-report inventory, which included four components: neurophysiological (item 1, 3, 6, 8, 12, 13, 19), autonomic (item 2, 18, 20, 21), subjective (item 4, 5, 9, 10, 14, 17), and panic (item 7,11,15,16) symptoms [28].

### 2.5. Procedure

Research data were collected in two stages. The participants participated in a face-to-face information exchange. Two trained research assistants explained the research aims and procedures and the participants signed the informed consent. All the study participants completed a questionnaire (the Beck Anxiety Inventory (BAI)) at the pre- and post-practicum stages (to screen whether their BAI score was ≥8) and a qualitative questionnaire (the open-ended questions) post-practicum.

The convenience and privacy of the setting were considered for the safety of the participants. Primarily, the investigator selected the students’ practicum centers as the study location and investigated the students without the involvement of any other students or nursing staff. To express our gratitude to the participants in this study and to make them feel more willing to participate, each participant was rewarded a coupon (NTD 50) after completing the questionnaire.

### 2.6. Ethical Considerations

Ethical permission was obtained from the Institutional Review Board of Kuang Tien General Hospital (people no. 10145). Before completing the questionnaires, participants offered both oral and written consent, were informed of the aims of the study and how the collected data would be analyzed, and were assured of absolute confidentiality. The convenience and privacy of the setting were considered for the participants’ safety. Primarily, the investigator chose the students’ practicum centers as the study location and investigated them without the involvement of other student members.

### 2.7. Data Analysis

Quantitative data used SIBM SPSS Statistics 27.0 (in HungKuang University, Taichung, Taiwan) by way of percentage, frequency, mean, and standard deviation to describe participants’ demographics and anxiety condition. 

By performing a content analysis on the qualitative questionnaire material, the practicum-related anxiety experience of the participants regarding their clinical nursing practice was thoroughly analyzed. We selected one piece of qualitative data for analysis and formed a basic database. In the second step, new qualitative data were continuously and repeatedly added to the database until the data characteristics became saturated and stabilized [29]. In the third step, each characteristic was defined and named as a sub-theme. In the final step, several sub-themes were put together to form one overarching theme.

### 2.8. Rigor for Qualitative Data

The study rigor was evaluated. Researchers collaborated with the qualitative advisor to increase the theoretical sensitivity of the study. The trustworthiness of the research data was established through four criteria [30,31]. First, to achieve credibility, the data were written down on the questionnaires by all participants to establish the authenticity and trustworthiness of the research data. Second, to achieve transferability, we selected nursing students with various experiences, such as students from different settings or units. Few controlling factors were used and we analyzed a large volume of data. Third, we ensured the trustworthiness of data analysis by peer-review using two researchers and constantly recoded the research data to reach an agreement. Finally, research data were collected to ensure they were sufficiently saturated to attain an accurate reflection of all themes by using two researchers to read and analyze the raw data and carefully ensure the participants’ experiences were heard and understood.

## 3. Results

We recruited 37 female nursing students. The mean age of participants was 20.7 years old. The demographics of participants are shown in Table 1. 

The mean post-practicum score was 11.62 (6.13) with the 4-year BSN being the highest scoring group. For the BAI components, post-practicum BAI scores were changed the most in the neurophysiological and panic components. For the practicum anxiety symptoms, post-practicum BAI scores were changed the most in the bad sleep symptom. For the coping strategies, post-practicum BAI scores were changed the most in the self-adaptation category for nursing students. Post-practicum BAI scores were decreased in those who used coping with teachers’ help. 

After analyzing the content of open questions, we conducted a conceptual structure of these study findings (Figure 1). ‘*Bad sleep and fatigue accompanies the nursing clinical practicum*’ is the core dimension. 

As the antecedent condition, students *encountered anxiety-inducing situations in clinical settings,* including one of three situations: (1) difficulty adapting to the medical team, such as teaching/communicating/interacting with patients and families and communicating with/interacting with/being taught by medical group members (e.g., doctors, advising nurses, or clinical teachers); (2) difficulty performing nursing processes, such as writing case evaluations or case study homework and facing unfamiliar patients/teachers; (3) difficulty performing nursing skills, such as nursing skills and dealing with critical situations or emergencies.

*Being nervous and scared* indicated that the students assessed these situations as stressful conditions. They started being nervous/afraid/worried/anxious/scared, not knowing how to care/showing no expression, being afraid of not being able to finish assignments and the workload in clinical practice, and being afraid of not being able to prepare well for patient care in clinical practice.

*Coping through self-adaptation* and *coping through teachers’ help* indicated that students tried coping through these stressful situations. They would begin the process of *coping through self-adaptation* by actions such as preparing materials, becoming familiar with the persons or procedures at work, sleeping early, and deep breathing. Furthermore, *coping through teachers’ help* involved gaining positive resources from teachers and clinical advisors, such as standing by their side when teaching, embodying non-stressful mannerisms, offering enough time or individual space for reflection, and potentially helping with and sharing in students’ workloads. This showed that students had positive feelings toward their clinical practicum even though they experienced anxiety during the period.

In cases of coping failure, students would *have a bad sleep in the night* and then be *tired and fatigued all day* and this could become a reinforcing cycle with *being nervous and scared*. This conceptual model shows that students experienced bad sleep and fatigue as symptoms of anxiety during the clinical practicum.

An expression of each theme is presented to illustrate the study findings.

Theme 1: Encountered Anxiety-Inducing Situations in Clinical Settings

This group of nursing students felt stress regarding their professional competences, such as difficulty adapting to the medical team, difficulty performing nursing processes, and difficulty performing nursing skills.

Sub-theme 1: Difficulty Adapting to the Medical Team

This group of nursing students felt stress regarding these professional competences: “teaching, communicating, or interacting with patients or families”, “communicating with, interacting with, or being taught by medical group members (e.g., doctors, advising nurses or clinical teachers)”.

Teaching, communicating, or interacting with patients and families

These participants found that they could not complete their nursing work well and could not understand what the patient meant. For instance, one nursing student said: “I couldn’t understand what he meant when patient spoke in Hokkien!” (No. 41). Another student said, “owing to the bad attitude of my patient, we couldn’t keep a good relation-ship!” (No. 58). The third student said, “the families still keep in pushing me for my patients’ state analgesics although I had explained that we were waiting for the drugs!” (No. 59). The fourth student said, “On facing patients, families, teachers, and nurses, I felt nervous and heart beat increased!” (No. 62). The fifth student said, “when I was teaching the patients and their families!” (No. 79). The sixth student said, “Some patient had poorly emotional control management may conflict with others.” (No. 96). The seventh answerer said, “didn’t know how to communicate with the patient, I didn’t know what could I do!” (No. 144). The eighth student said, “it made me crazy when patients didn’t touch me respectfully!” (No. 145). The ninth student said, “it was hard to explain the objects of what I was willing to perform to those whom couldn’t speak or understand Mandarin!” (No. 228).

2.Communicating with, interacting with, or being taught by medical group members (e.g., doctors, advising nurses, or clinical teachers)

These participants thought that they struggled with group education. For instance, one answerer said: “I must read many and work hard when I was off duty at home, …when I was asked some questions and couldn’t answer correctly!” (No. 5). Another answerer said, “I was worry about writing down nursing recording incorrectly! My actions were too slow to accompany with and were always delayed the time schedule of the clinical nurses!” (No. 6). Another answerer said, “Communicating with clinical nurses! Some were nice but some were not!” (No. 23). The fourth answerer said, “When I could not answer the questions from school teachers!” (No. 34). The fifth answerer said, “On facing patients, families, teachers, and nurses, I felt nervous and heart beat increased!” (No. 62). The sixth answerer said, “I felt anxious when I was communicating with other medicine-group members!” (No. 76). The seventh answerer said, “In the moment of oral presentation and tests by the teacher!” (No. 118). The eighth answerer said, “The everyday morning meeting! Some-times, too many things need take home to prepare, resulted in having no time to rest enough!” (No. 119). The ninth answerer said, “When I was reading and translating the contents of a chart in English to Chinese!” (No. 131). The tenth answerer said, “I was changed to be nervous in the situations when I must to write the recordings according to my nursing works to my patients!” (No. 144). The eleventh answerer said, “The feeling of taking a clinical test was very anxious! To take a test in clinical practicum is important, because it can push us to study more exactly. But I was so tired after all day long practicum and many studies were taking to home. I was slept so late, tire to practice in another day accomplished with tests!” (No. 155).

Sub-theme 2: Difficulty Performing Nursing Processes

This group of nursing students felt stress regarding professional competences, such as “writing case evaluations or case study homework” and “facing unfamiliar patients”.

1.Writing case evaluation or case study homework

These participants felt that establishing therapeutic relationships or having therapeutic conversations with patients and their families was difficult. For instance, one student said: “I was afraid of data collecting for client-analysis was incomplete and could not apply knowledge learning about well!” (No. 6). Another student said, “I worry about my home-work was not done as expected!” (No. 58).

2.Facing unfamiliar patients

These participants struggled when they had to collect data surrounding the nursing process, especially for nursing evaluations. For instance, one answerer said: “Every time when making a health-teaching, I was becoming nervous and afraid of being not so perfect in playing a nursing profession role in order to gain my patients’ confident and trust!” (No. 134). 

Sub-theme 3: Difficulty Performing Nursing Skills

This group of nursing students felt stress regarding professional competences, such as “nursing skills” and “dealing with critical or emergency conditions of patients”.

1.Nursing skills

These participants stated that they were always afraid of doing something wrong when completing nursing−recording paperwork. For instance, one answerer said: I was very serious about not doing the skill well when operating a penetrating skill on a patient!” (No. 26). Another answerer said, “I was afraid of failure when taking a blood test for my patient!” (No. 216). The third answerer said, “I feel anxious in the situation of preparing on Foley!” (No. 58). The fourth answerer said, “the first time on doing a penetrating technique!” (No. 62). The fifth answerer said, “before performing a skill” (No. 79). The sixth answerer said, “I was serious in taking a skill. I felt my hands were tremulous so that I must to take a deep breath to relax myself!” (No. 102). The seventh answerer said, “The moment when I was facing the unfamiliar skills and routine procedures like of per-forming the admission of a new patient!” (No. 110). The eighth answerer said, “On case of the things I had never seen such as unfamiliar routines, procedures, heath teachings, nursing cares!” (No. 118). The ninth answerer said, “I was very serious about not doing the skill well when operating a penetrating skill on a patient!” (No. 134). The tenth answerer said, “Foley catheterization and Foley care! Female patient! It was not smooth when inserting the end of the catheter into patients’ urethra! The family was in aside! I felt a lot of stress!” (No. 155). The eleventh answerer said, “I feel anxious in the situation of preparing for on Foley catheter!” (No. 210).

2.Dealing with critical or emergency conditions of patients

When caring for unstable or emergency patients and dealing with untreated patients, students felt uneasy. For instance, one answerer said: “The managements of emergence! Worry about myself couldn’t deal with the patient’s condition!” (No. 110).

Theme 2: Being Nervous and Scared

This group of nursing students considered themselves anxious, expressing feelings such as nervousness and fear, being scared/not knowing how to care/showing no expression, being afraid of/could not finish assignments and the workload in clinical practice, and being afraid of/could not prepare well for patient care in clinical practice.

Sub-theme 1: Nervous/afraid/worried/anxious/scared

These participants stated that they were always afraid of doing something wrong when experiencing the anxious event. For instance, one answerer said: “I was very serious about not doing the skill well when operating a penetrating skill on a patient!” (No. 26). Another answerer said, “I was worry if patient acknowledge of my student-identification then they would never trust me!” (No. 228). The third answerer said, “I felt anxious in the first day to care a new patient. I felt anxious when I wrote the case study…, it is really hard for me, it is really let me felt anxious!” (No. 63). The fourth answerer said, “I was worry about whether families trust me or not!” (No. 144). The fifth answerer said, “I felt anxious when doing everything in the ward during clinical practicum!” (No. 182).

Sub-theme 2: Not knowing how to care/not showing expression

These participants were scared in the practice environment. For instance, one answerer said: “I had never encountered such client, I didn’t know how to express and how to care in the first time!” (No. 5). Another answerer said, “facing to the psychosis patient, I was scared sometimes because of their poor self-control ability!” (No. 189). 

Sub-theme 3: Being afraid of/could not finish assignments and workload in clinical practice

These participants struggled to prepare themselves for health teaching. For instance, one student said: “Morning meeting had taken too much time, so I was afraid of being too late or preparing enough to take care of my patients!” (No. 118). Another student said, “Morning meeting was too long so as to doing my jobs too late!” (No. 119). Another nursing student said, “I was afraid of doing something wrong or could not finish my works!” (No. 102).

Sub-theme 4: Being afraid of/could not prepare well for patient care in clinical practice

These participants struggled to prepare themselves for health teaching. For instance, one student said: “Morning meeting had taken too much time, so I was afraid of being too late or preparing enough to take care of my patients!” (No. 118).

Theme 3: Coping through Self-Adaption

This group of nursing students tried to use particular mannerisms to cope with stress and adjust themselves when they felt stressors, such as by preparing materials, becoming familiar with the environment/persons/work/materials, sleeping early, and deep breathing. 

Sub-theme 1: Preparing materials 

These participants always prepared themselves as best as they could. For instance, one answerer said: “I was prepared more and more…!” (No. 5).

Sub-theme 2: Becoming familiar with the environment, persons, work, and materials

These participants always tried to familiarize themselves with or adapt to changes in the workplace. For instance, one answerer said: “I felt better when I was familiar with the environment, person, every things and materials in the workplace!” (No. 62).

Sub-theme 3: Sleeping early

These participants went to sleep early. For instance, one answerer said: “I just go to sleep very early!” (No. 59).

Sub-theme 4: Deep breathing

These participants need to take a deep breath to adjust themselves. For instance, one answerer said: I must to take a deep breathing so as to feel of relaxing!” (No. 59).

Theme 4: Coping through Teachers’ Help

This group of nursing students asked for helpful resources from teachers in clinical settings, such as standing by their side when teaching, embodying non-stressful mannerisms, offering enough time or individual space to think, and potentially helping with and sharing in students’ workloads.

Sub-theme 1: Standing by their side when teaching 

These participants felt that it was beneficial when their teachers or nurses stood by their side when teaching. For instance, one answerer said: “I was very serious about not doing the skill well when operating a penetrating skill on a patient!” (No. 166). Another answerer said, “I felt anxious when doing everything in the word during clinical practicum!” (No. 208). The third answerer said, “The moment when I was facing the unfamiliar skills and routine procedures like of performing the admission of a new patient!” (No. 122). The fourth answerer said, “The moment when I was facing the unfamiliar skills and routine procedures like of performing the admission of a new patient!” (No. 96). The fifth answerer said, “The moment when I was facing the unfamiliar skills and routine procedures like of performing the admission of a new patient!” (No. 67). The sixth answerer said, “The moment when I was facing the unfamiliar skills and routine procedures like of performing the admission of a new patient!” (No. 166). The seventh answerer said, “The moment when I was facing the unfamiliar skills and routine procedures like of performing the ad-mission of a new patient!” (No. 209). The eighth answerer said, “The moment when I was facing the unfamiliar skills and routine procedures like of performing the admission of a new patient!” (No. 58). The ninth answerer said, “The moment when I was facing the un-familiar skills and routine procedures like of performing the admission of a new patient!” (No. 153).

Sub-theme 2: Non-stressful mannerisms 

These participants felt that it was beneficial when their teachers or nurses had non-stressful mannerisms when teaching. For instance, one answerer said: “I was very serious about not doing the skill well when operating a penetrating skill on a patient!” (No. 134). Another answerer said, “I felt anxious when doing everything in the word during clinical practicum!” (No. 189). The third answerer said, “I feel anxious in the situation of preparing on Foley!” (No. 230). The fourth answerer said, “the first time on doing a penetrating technique!” (No. 183). The fifth answerer said, “the first time on doing a penetrating technique!” (No. 122). The sixth answerer said, “the first time on doing a penetrating technique!” (No. 23). The seventh answerer said, “The moment when I was facing the unfamiliar skills and routine procedures like of performing the admission of a new patient!” (No. 6). The eighth answerer said, “The moment when I was facing the unfamiliar skills and routine procedures like of performing the admission of a new patient!” (No. 59). The ninth answerer said, “The moment when I was facing the unfamiliar skills and routine procedures like of performing the admission of a new patient!” (No. 144).

Sub-theme 3: Offering enough time or individual space to think 

These participants felt that it was beneficial when their teachers or nurses gave them enough time or individual space to think. For instance, one answerer said: “I was very serious about not doing the skill well when operating a penetrating skill on a patient!” (No. 119). Another answerer said, “I felt anxious when doing everything in the word during clinical practicum!” (No. 182). The third answerer said, “I feel anxious in the situation of preparing on Foley!” (No. 134). The fourth answerer said, “the first time on doing a penetrating technique!” (No. 206). The fifth answerer said, “before performing a skill” (No. 119). The sixth answerer said, “I was serious in taking a skill. I felt my hands were tremulous so that I must to take a deep breath to relax myself!” (No. 58). The seventh answerer said, “I was serious in taking a skill. I felt my hands were tremulous so that I must to take a deep breath to relax myself!” (No. 211).

Sub-theme 4: Helping with and sharing in students’ workloads 

These participants felt that it was beneficial when their teachers or nurses helped with and shared the students’ workloads. For instance, one answerer said: “I was very se-rious about not doing the skill well when operating a penetrating skill on a patient!” (No. 122). Another answerer said, “I felt anxious when doing everything in the word dur-ing clinical practicum!” (No. 210).

Theme 5: Have a Bad Sleep in the Night

These participants thought that they struggled to engage in their nursing shift reporting. For instance, one nursing student said: “I have to awake early to avoid delay on schedule…, I was always afraid of traffic accident in my way to hospital by riding motorcycle!” (No. 41). Another nursing student said, “sleeping late, one all day again,!” (No. 155). 

Theme 6: Tired and Fatigued all Day

These participants thought that they struggled to engage in their nursing shift reporting. For instance, one nursing student said: “I deed sleep early, but felt very tired every day!” (No. 59). Another answerer said, “I was troubled by the same things. I was sleeping early in the night, but I still felt tired when waking!” (No. 102). The third answerer said, “I was very tired!” (No. 155).

## 4. Discussion

### 4.1. Bad Sleep and Fatigue Accompanied Students in the Nursing Clinical Practicum

Most of the anxiety students experienced had resulted in subjective symptoms in their daily life. These included feeling “nervous and afraid”, “having a bad sleep in the night”, and feeling “tired and fatigued”.

What conditions will induce these chief subjective symptoms in anxious nursing students? There are few studies focused on the chief complaints and subjective symptoms reported by nursing students. In addition, it is unknown what conditions induced by these chief complaints and subjective symptoms might occur in these nursing students.

Fatigue was a major symptom of low-grade systemic inflammation. In the study by Pitsavos, Panagiotakos, Papageorgiou, Tsetsekou, Soldatos, and Stefanadis [32], they pointed out “anxiety in relation to inflammation and coagulation markers, among healthy adults”. Nowadays, more and more studies state that anxiety is related to inflammation in many health disturbances, including mental and physical health problems. For example, Vogelzangs, Beekman, de Jonge, and Penninx [33], in a large adult cohort study, examined 18–65-year-old persons with a current or remitted anxiety disorder for the association between anxiety disorders and anxiety characteristics with several inflammatory markers. They made the conclusion that low-grade systemic inflammation was present in people with anxiety disorders. In their study, they suggested alternative treatments could possibly benefit anxiety patients. Yang et al. [34] revealed that systemic inflammation may induce anxiety in mice. Furthermore, Martin et al. [35], in a population-based cohort study conducted from 2002–2010 including a total of 730 men aged 35–80 years, found that CRP and TNF-α have significant moderating effects on the development of anxiety. In addition, Song et al. [36] suggested that systemic inflammation is associated with the prevalence of anxiety symptoms in glioma patients. Another study by Michopoulos et al. [37] reviewed studies and indicated that anxiety disorders may be related to heightened pro-inflammatory markers and targeting inflammation may be used to treat fear- and anxiety-based disorders in the future.

### 4.2. Encountering Anxiety-Inducing Situations in Clinical Setting Made Nursing Students Feel Nervous and Scared

In the last 10 years, many studies have pointed out that student nurses experience a series of anxiety-inducing situations: talking to patients, talking to the patient’s family, reporting to the team leader, preceptor, or charge nurse, talking with physicians, asking questions of the faculty, being evaluated by the faculty, patient teaching, procedures, hospital equipment, fear of making mistakes, patient morning care, availability of instructors, the initial clinical experience on a unit, anticipatory in-hospital preparation, being observed by instructors, being late, fear of making mistakes, and being observed by instructors. In addition, during the orientation period, nursing students had to follow and learn from the nurses. They had to be “communicating with clinical nurses!” and found of the nurses: “some were nice but some were not!”. Students beginning to interact with patients and families may feel “somewhat anxious” because they “didn’t know the answers of my patients’ and families’ questions!” [5,7,37,38]. 

Nursing students felt nervous and afraid, described as being nervous/afraid/worried/anxious, being scared/not knowing how to care/not showing expression, being afraid of/not being able to finish assignments and the workload in clinical practice, being afraid of/ not being able to prepare well for patient care in clinical practice. This is consistent with the study of Graj et al. [8] regarding ‘the worrying trend’, which discusses the faculty’s evaluation process as an important cause of stressful relationships between student nurses and the faculty. However, clinical tests are a major consideration in objective evaluations to confirm students’ learning outcomes. If the faculty does not correctly evaluate students, they may feel they have been done a disservice [11]. A positive work environment with personalization is an important element for a positive work environment [39]. In addition, as the study by Tang [6] mentioned, encountering a variety of patients with different illnesses is highly challenging for nursing students. The conflict created by the differences between the actual and expected illness manifestations can cause anxiety in student nurses. For example, facing ‘patient death’ is a specific clinical anxiety situation for nursing students [5].

As mentioned by Baluwa et al. [40], the major contributors of nursing students’ clinical practicum stress came from lecturers, clinical teachers, and nursing staff.

Liu et al. [18] studied and reported that the most common source of stress was the need for knowledge and skills.

### 4.3. Coping Style of Nursing Students: Self-Adaptation and Teachers’ Help

Coping is proposed as the key to maintaining well-being and a satisfactory performance. Recent definitions view stress as arising from the interaction between the person and the situation. There are two functions of coping—dealing with a problem that has arisen (problem-focused coping) and regulating associated emotions (emotion-focused coping). Different coping approaches involve regulating emotions in a positive or negative way [41].

In this study, students tried to apply resources to cope with and adjust to anxiety-inducing conditions, including coping through self-adaptation and coping through teachers’ help. 

Common anxiety coping strategies for students, as mentioned by Baluwa et al. [40], involved active coping and planning. Sucuogluet et al. [42] showed that relaxation techniques and positive self-talk are the most common methods of managing test anxiety. Ortega et al. [43] suggested that cognitive behavioral therapy and progressive muscle relaxation programs are useful for anxiety management in clinical practice nursing students. Vurara et al. [44] founded emotional freedom techniques (EFT), which significantly reduced exam anxiety and other anxiety indicators. 

In this study, students used four kinds of mannerisms to help them cope in anxiety-inducing situations, such as preparing materials, becoming familiar with the environment/persons/work/materials, sleeping early, and deep breathing. These were consistent with studies by Cornine [45] and Aloufi, Jarden, Gerdtz, and Kapp [46], who discussed ways of decreasing the anxiety of nursing students during clinical practicum, which were considered positive coping mechanisms [41].

The faculty usually designed a practicum plan for student nurses according to the time−series orientation in the first week and functional care in the second week, followed by the total care of one or two patients. In the orientation period, nursing students may begin to experience “the morning meeting in every day!” and “felt anxious when doing everything in the word”. This feeling persists “during clinical practicum!” They “have to awake early to avoid delay”, causing students to “always awaken in the mid-night!” Every morning, the students ride motorcycles and are always “afraid of traffic accident” on the way to hospital.

Anxiety coping strategies for teachers to help students was mentioned by Kennya et al. [47]; clinical faculties may consider integrating mobile technology into students’ clinical learning experiences to help reduce their anxiety. A study by Deryaet et al. [48] suggested a web-based software system that can positively affect the motivation and time management of midwifery nursing students. Shahsavari et al. [49] indicated that students may achieve lower anxiety levels, higher levels of clinical self-efficacy, and better clinical skills during their internships by participating in a refresher course. Kachaturoff et al. [50] suggested that teachers use peer mentoring to decrease students’ anxiety. Ioannou et al. [51] suggested teachers to apply virtual reality (VR) to act as an emotion-focused distraction intervention for the anxiety and fatigue symptoms of nursing students.

In this study, teachers thought four kinds of coping strategies would be helpful to the students. They were: (1) stand by their side when teaching; (2) employ non-stressful mannerisms; (3) offer enough time or individual space to think; (4) help with and share in students’ workloads. These were consistent with Devi [52], who found that resilience is a crucial mediator of clinical practice-related anxiety in nursing students and suggested teachers keep a well-balanced relationship between academic demands and students’ private life to help students counteract anxiety. Savitsky et al. [53] mentioned that nursing staff may contribute a stable educational framework, providing high-quality distant teaching and encouraging and supporting students to overcome their challenges. 

Regarding the issue of coping strategies, although there were many strategies that could be used by students, the study of Baluwa et al. [40] warned that some students began coping with substance use because they were unable to cope with high-level stress. Therefore, in this study, we studied the students who are high in anxiety, meaning that the participants were a group that may face failure of coping strategies and highly stressful situations. In these cases, we need to pay attention to avoid substance use.

In the above context, it is noteworthy that a bad sleep, feeling tired, and symptoms of fatigue appeared in nursing students. In anxiety reactions, ‘bad sleep’ is a major cause of low-grade systemic inflammation [54] and ‘fatigue’ is also a related symptom of low-grade systemic inflammation [55]. According to Pitsavos et al. [32], Vogelzang et al. [33], Yang et al. [34], Martin et al. [35], Song et al. [36], and Michopoulos et al. [37], anxious people may have low-grade systemic inflammation. 

The inability of nursing students to cope with stress might induce nervousness and anxiety and that anxiety might interrupt sleep; also, bad sleep was related to fatigue, leading to an effect on daily performance [12,13]. Therefore, nursing teachers need to pay attention to these students to avoid the risk of and evaluate inflammation symptoms. 

Combining the above results, we can see that in the past 10 years, most research did not explore the symptoms of anxiety nor point out the characteristics of anxiety symptoms. Are the fatigue symptoms of practicum-related anxiety similar to low-grade systemic inflammation? Are there are any differences in severity over time? Could the data analyzed from qualitative data and quantitative data be the same? 

## 5. Limitations and Strengths

A major strength of the current study was its qualitative design, which aimed to explore the actual situations that participants encountered. Furthermore, the current study created a conceptual model of practicum-related anxiety symptoms in female nursing students. There were several limitations in this study. First, the sample size was small, which may prohibit broader conclusions. In addition, the participants in this study were nursing students from only one educational institution and their experience was only researched at one stage of their clinical practicum. Thus, the ability to generalize these findings to a wider population is limited and there were potential issues with reliability. However, many of the results of this study echo the findings of previous literature. This may support this study’s reliability.

## 6. Conclusions

The findings indicate that feeling “nervous and afraid of”, “having a bad sleep in the night”, and feeling “tired and fatigued all day long” are critical anxious emotional reactions in female nursing students. Teachers should deal with anxious emotional reactions and monitor students’ sleep conditions to improve students’ tiredness and fatigue symptoms. Training nursing students to cope through self-adaptation and teachers’ help is suggested. We suggest that the faculty could assist in the training of students through case analyses by actively asking questions and sharing their experience to help clinical students overcome their anxiety. In the school curriculum, students could spend more time learning communication skills or participating in extra courses. In their personal lives, students may receive help from their families or friends to work to better apply time-management skills. Regarding the language use in Hokkien, students can prepare through daily life communication.

## Figures and Tables

**Figure 1 ijerph-19-07479-f001:**
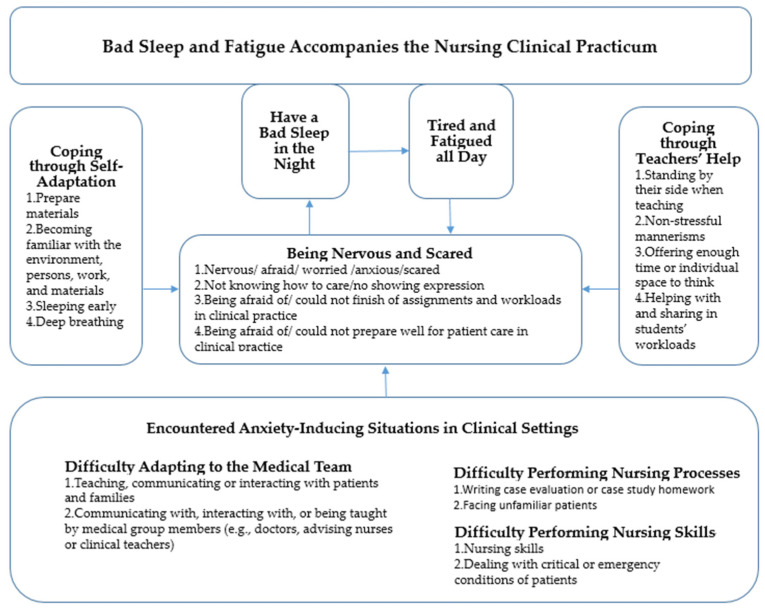
A conceptual model of practicum-related anxiety symptoms in female nursing students.

**Table 1 ijerph-19-07479-t001:** Demographic distribution of participants.

Variables	*n* (%)	Pre-Practicum M (S.D.)	Post-Practicum M (S.D.)
Age	37 (100)	10.21 (6.87)	11.62 (6.13)
Nursing program			
4-year BSN	8 (21.6)	13.88 (8.10)	13.63 (4.50)
2-year BSN	13 (35.1)	10.15 (6.93)	10.31 (5.60)
5-year ABSN	16 (43.2)	8.69 (5.88)	11.69 (7.22)
Practice setting			
Med. and sur.	18 (48)	8.39 (6.00)	12.06 (7.27)
Obs.	8 (21.6)	11.38 (7.33)	11.00 (7.17)
Ped.	2 (5.4)	9.50 (0.71)	9.50 (2.12)
Psy.	4 (10.8)	15.00 (8.37)	9.50 (1.29)
Com.	5 (13.5)	12.20 (8.90)	13.60 (2.88)
BAI components (pre)			
Neurophysiological symptoms	30 (81.1)	11.93 (6.54)	10.33 (4.89)
Autonomic symptoms	34 (91.9)	11.15 (6.54)	11.71 (6.39)
Subjective symptoms	24 (64.9)	13.50 (6.28)	10.46 (5.48)
Panic symptoms	32 (86.5)	11.28 (6.75)	11.69 (6.50)
BAI components (post)			
Neurophysiological symptoms	34 (91.9)	9.82 (6.86)	12.32 (5.69)
Autonomic symptoms	35 (94.6)	10.14 (7.01)	12.29 (5.61)
Subjective symptoms	26 (70.3)	10.65 (7.53)	13.35 (6.14)
Panic symptoms	34 (91.9)	9.68 (6.54)	12.35 (5.68)
Practicum-related anxiety symptom	13 (35.1)	10.62 (7.70)	11.23 (6.33)
Fatigue	3 (8.1)	7.67 (2.52)	8.67 (1.15)
Bad sleep	2 (5.4)	2.50 (3.54)	9.00 (1.41)
Nervous and scared	10 (27.0)	12.30 (7.73)	11.80 (7.18)
Coping strategy used	23 (62.2)	13.13 (6.98)	12.13 (6.12)
Self-adaptation	3 (8.1)	6.33 (3.51)	11.00 (2.65)
Teachers’ help	21 (56.8)	13.95 (6.72)	12.19 (6.37)

## Data Availability

The data presented in this study are available on request from the corresponding author. The data are not publicly available due to privacy.

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
