# Peer review of "Coping with the Experience of Bad Sleep and Fatigue Associated with the Nursing Clinical Practicum"

_ijerph, 2022, doi:10.3390/ijerph19127479_

Round 1
Reviewer 1 Report
Congratulations for the manuscript. I think you reached satisfactory level.
Reviewer 2 Report
The article has been practically rewritten from scratch. There were important errors that have been resolved following the recommendations of the reviewers.
The article now meets the requirements to be published, but in my opinion, it is of low interest and originality.
This manuscript is a resubmission of an earlier submission. The following is a list of the peer review reports and author responses from that submission.
Round 1
Reviewer 1 Report
The title is extremely long and inaccurate. If statistics are applied, however basic and descriptive, the research methodology used will be mixed. A common mistake is to consider research as only qualitative because it deals with qualitative variables. Many of these can be quantified, and therefore statistics can be applied. This inevitably changes the design.
On the other hand, it should be noted that the data were collected 9 years ago, and are out of date. Nowadays, the global pandemic situation is a very obvious factor affecting the anxiety experienced by students. Furthermore, there is a clear gender bias, as there is not a single male participant. We know, that the nursing profession is mostly female, but it is a heterogeneous and mixed social group, with a significant male presence as well. This aspect should not have been overlooked because it is a bias in the results.
Finally, it should be noted that the bibliography is very old, too old.
Reviewer 2 Report
Dear authors,
the research topic is current and can be interesting. Grammar and English proofreading is needed throughout whole manuscript.
Abstract - do not use abbreviations in abstract. Methodology is confused. Conclusion is not clear.
Introduction - At he end of introduction you have the same conclusion as in abstract. You have to prove scientific contribution of your research, and to state what research gap you are aiming to fill.
For international audience of this journal it would be interesting to better describe what this study brings to international science.
Methodology -also confused, I do not understand use of secondary data? Maybe to clarify better . How you defined themes prior the questioning the participants or before. It is usual that themes comprise from the answers. How do you decide what was the sample, when saturation occurred?
Results - in result section you said 47 and in table you presented 55 students. Table 1 is not suitable, nor Figure 1.
Discussion - do you have study limitations, do you have suggestions for further research. Will
Conclusion - whats new?
Round 2
Reviewer 1 Report
I consider that none of the problems I indicated in my previous review report have been solved and that this article is not of sufficient quality to be published in IJERPH.
The title is still inadequate, I do not agree with the design as explained, there is a gender bias in this study; even if in the future we try to avoid it, there is a clear gender bias in this study. In the bibliography, although there is a part that is up to date, there is another part that is still outdated. The results are often not well expressed. It is not a question of stating verbatim what each of the participants answered, but rather of making a theoretical sampling of the discourses, extracting the most significant ones, abstracting to explain the fact, and contrasting it with the bibliography. This is not done correctly in the article. The new conclusions do not contribute to anything new or novel. I am very sorry for this.
Author Response
Point 1: The title is still inadequate,
Response 1:
We have change the title as”Coping with the Experience of Bad Sleep and Fatigue Accompanied in
Nursing Clinical Practicum”. Please see the revised manuscript in line 2-3.
Point 2: I do not agree with the design as explained
Response 2:
We changed the design of this study. Please see the revised manuscript in line 16-19, line 90-195.
Point 3: there is a gender bias in this study; even if in the future we try to avoid it, there is a clear gender
bias in this study.
Response 3:
To avoid gender bias, we changed the gender of participants to female. Please see the revised
manuscript in line 18, 86-89, 102- 104, line 113.
Point 4: In the bibliography, although there is a part that is up to date, there is another part that is still
outdated.
Response 4:
We had updated the bibliography. Please see the revised manuscript in line 635-758.
Point 5: The results are often not well expressed. It is not a question of stating verbatim what each of the
participants answered, but rather of making a theoretical sampling of the discourses, extracting the most
significant ones, abstracting to explain the fact, and contrasting it with the bibliography. This is not done
correctly in the article.
Response 5:
We rewrote the results and discussions and added the conceptual model (Figure 1). Please see the
revised manuscript in line 196-456, and line 458 –604.
Point 6: The new conclusions do not contribute to anything new or novel.
Response 6:
We changed the conclusions. Please see the revised manuscript in line 606-617.